# A 2D chiral microcavity based on apparent circular dichroism

Tzu-Ling Chen [1,2,6], Andrew Salij [3,6], Katherine A. Parrish [1,6], Julia K. Rasch [1], Francesco Zinna [4], Paige J. Brown [3], Gennaro Pescitelli [4], Francesco Urraci[4], Laura A. Aronica[4], Abitha Dhavamani [5], Michael S. Arnold [5], Michael R. Wasielewski [3], Lorenzo di Bari [4], Roel Tempelaar [3] ✉ & Randall H. Goldsmith [1] ✉

Engineering asymmetric transmission between left-handed and right-handed circularly polarized light in planar Fabry–Pérot (FP) microcavities would enable a variety of chiral light-matter phenomena, with applications in spintronics, polaritonics, and chiral lasing. Such symmetry breaking, however, generally requires Faraday rotators or nanofabricated polarization-preserving mirrors. We present a simple solution requiring no nanofabrication to induce asymmetric transmission in FP microcavities, preserving low mode volumes by embedding organic thin films exhibiting apparent circular dichroism (ACD); an optical phenomenon based on 2D chirality. Importantly, ACD interactions are opposite for counter-propagating light. Consequently, we demonstrated asymmetric transmission of cavity modes over an order of magnitude larger than that of the isolated thin film. Through circular dichroism spectroscopy, Mueller matrix ellipsometry, and simulation using theoretical scattering matrix methods, we characterize the spatial, spectral, and angular chiroptical responses of this 2D chiral microcavity.

Chirality, a fundamental property found in nature, has significant implications in many scientific and technological fields[1]. Chiral molecules and materials exhibit unique interactions with light, potentially enabling matter to discriminate between the spin angular momenta of photons. This feature has led to growing interest in quantum control of chiral degrees of freedom for quantum information science (QIS), resulting in the exploration of techniques for manipulating electron spins[2], constructing spin filters[3], observing the chiral Purcell effect[4], developing chiral-induced spin selectivity[5], and developing quantum information storage and processing[6,7]. Moreover, the advancement of circularly polarized luminescent materials shows the ability to generate circular polarized light, leading to promising applications, including displays, optical sensing, and optical data storage[8].

Efficient transduction of circular polarizations between light and matter requires both strong light-matter coupling and high discrimination between left-handed and right-handed circularly polarized (LCP and RCP) light. In recent years, planar Fabry–Pérot (FP) microcavities with small mode volumes have received increasing attention as a straightforward means of reaching the strong-coupling regime by allowing multiple round trips between two mirrors in the resonant modes of the cavity, resulting in a longer effective interaction length of light with the embedded material. Through this mechanism, FP microcavities support photon-matter hybrid excitations known as polaritons, with diverse applications in condensed matter physics, quantum materials, and chemical reaction dynamics[9–13]. While FP cavities can sustain modes featuring well-defined circular motion of

[1]Department of Chemistry, University of Wisconsin-Madison, 1101 University Ave, Madison, WI 53706, USA. [2]Department of Photonics, National Yang Ming Chiao Tung University, 1001 Ta-Hsueh Road, Hsinchu, Taiwan. [3]Department of Chemistry, Northwestern University, 2145 Sheridan Rd, Evanston, IL 60208, USA. [4]Dipartimento di Chimica e Chimica Industriale, Università di Pisa, Via Giuseppe Moruzzi, 13, Pisa, PI 56124, Italy. [5]Department of Materials Science and Engineering, University of Wisconsin-Madison, 1415 Engineering Drive, Madison, WI 53706, USA. [6]These authors contributed equally: Tzu-Ling Chen, Andrew Salij, Katherine A. Parrish. ✉e-mail: roel.tempelaar@northwestern.edu; rhg@chem.wisc.edu

local field amplitudes[14], and such angular motion can in principle induce angular momentum states in non-centrosymmetric samples such as monolayer transition-metal dichalcogenides[15], standard FP cavities by themselves do not normally discriminate between circularly polarized optical modes. FP cavities would be similarly indiscriminate to circularly polarized optical modes even with a sample exhibiting natural optical activity embedded inside, because the handedness of circularly polarized light is reversed upon each reflection. However, microcavities that can offer asymmetric responses to LCP and RCP light enable access to important new operational capacities in QIS, including the creation of chiral polaritons that can be selectively initialized and accessed by circularly polarized light[16]. Consequently, there is a significant interest in breaking the chiral symmetry of FP microcavities. Here, we describe an accessible strategy for producing planar FP microcavities that exhibit asymmetric transmission[17] to circularly polarized light. The property originates in the 2D chirality of an embedded molecular thin film, and as a consequence, we refer to the cavity as a 2D chiral cavity. We note that this 2D chiral cavity is distinct from a 3D chiral cavity produced with polarization-preserving mirrors[18], as further discussed below.

A 2D chiral cavity can be created by including materials exhibiting a property called apparent circular dichroism (ACD), originating from the material's 2D chirality[19]. Materials possess 2D chirality when a system embedded in a plane becomes its mirror image when flipped, potentially leading to an ACD response that displays signal inversion upon sample flipping[19], Fig. 1. In contrast with natural optical activity, ACD allows for asymmetric transmission, which offers the opportunity to induce directionally dependent polarization properties and overcome the critical limitation in microcavities that prevents amplification. New options for creating 2D chiral microcavities can derive from novel materials featuring chiral π-conjugated oligothiophenes that demonstrate ACD. During the fabrication process, the oligothiophenes

form thin films with a preferential orientation on a substrate as the chiral conjugated side chains anneal and self-assemble into ordered crystals which exhibit chiroptical responses of varying sign and strength[20]. These films demonstrate asymmetric transmission: signal inversion of the handedness of the circularly polarized component preferentially absorbed by the two opposite faces of the sample[21,22], as shown in Fig. 1. This inversion is the signature of ACD, which does not intrinsically originate at the molecular or microscopic level as in the case of natural optical activity, but rather originates from macroscopic ordering due to nonparallel axes of LD and LB, which occurs in oriented, low symmetry systems[23–25]. Crystalline thin films necessarily possess the requisite orientation, and the presence in them of organic chromophores with multiple electronic transitions oriented in different directions ensures that there is a selective phase shift followed by diattenuation with a different principal axis[25]. When the projection of multiple electronic transition dipoles (e.g., in Fig. 1a) in crystals overlap in energy while forming a 2D chiral object, ACD occurs near the electronic absorption band[25]. While the molecules responsible for ACD-active films may have chiral chemical structures, the ACD response is fundamentally 2D chiral, occurring due to offset optical axes whose mutual orientation changes sign upon sample flipping.

In this communication, we show how ACD in organic molecules, a 2D chiral phenomenon, offers the ability to easily engineer asymmetric transmission in planar FP microcavities. Thus, a planar FP microcavity, already straightforward to produce, can be easily converted to a 2D chiral microcavity by simply spin coating an organic thin film onto one of the mirrors before fabrication. Importantly, such a process does not require difficult electron-beam lithography or other nanofabrication techniques, making it uniquely capable of wide applicability. Despite its potential for conveniently breaking the symmetry between chiral modes in a FP microcavity, ACD has yet to be used to such an effect. Here, we demonstrate the usage of ACD to alter the chiroptical

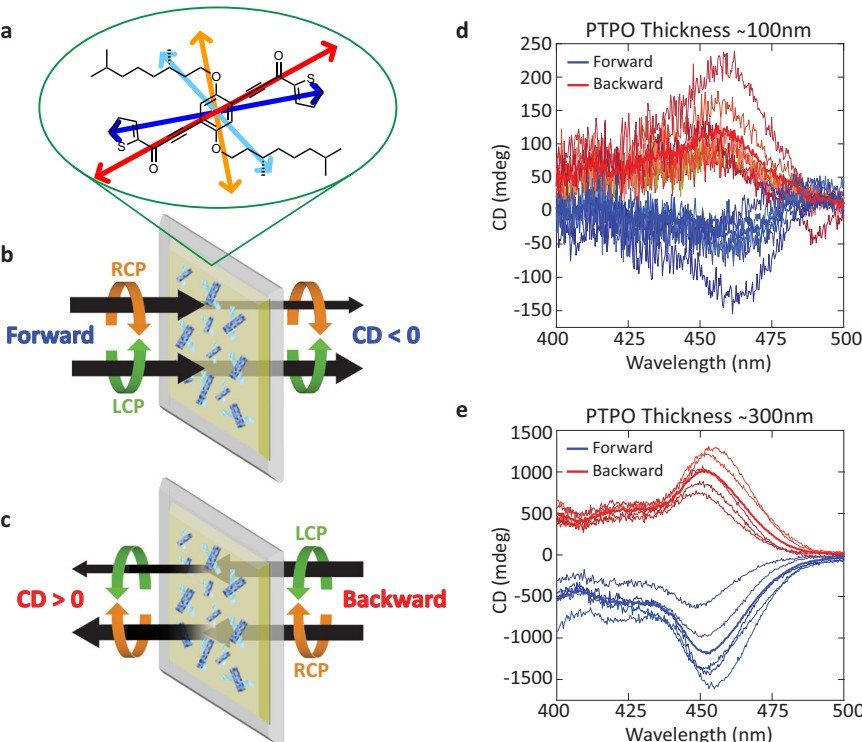

**Fig. 1 | Chiroptical properties and signal inversion of ACD-exhibiting PTPO films. a** Structure of PTPO with sketch of four largest electronic transition dipoles used in modeling colored. Asymmetric transmission properties of a 2D chiral thin film in the (**b**) forward geometry of a PTPO film demonstrating increased transmission of LCP light over RCP light, and in a (**c**) backward geometry where the

sign of the CD signal is reversed. Chiroptical properties of a 100 nm (**d**) and 300 nm (**e**) PTPO film on an HR substrate measured at different spatial locations. Blue curves are acquired in the forward geometry (see **b**) and red curves are acquired in the backward geometry. The thick solid curves show the average from several different spots in the backward or forward direction.

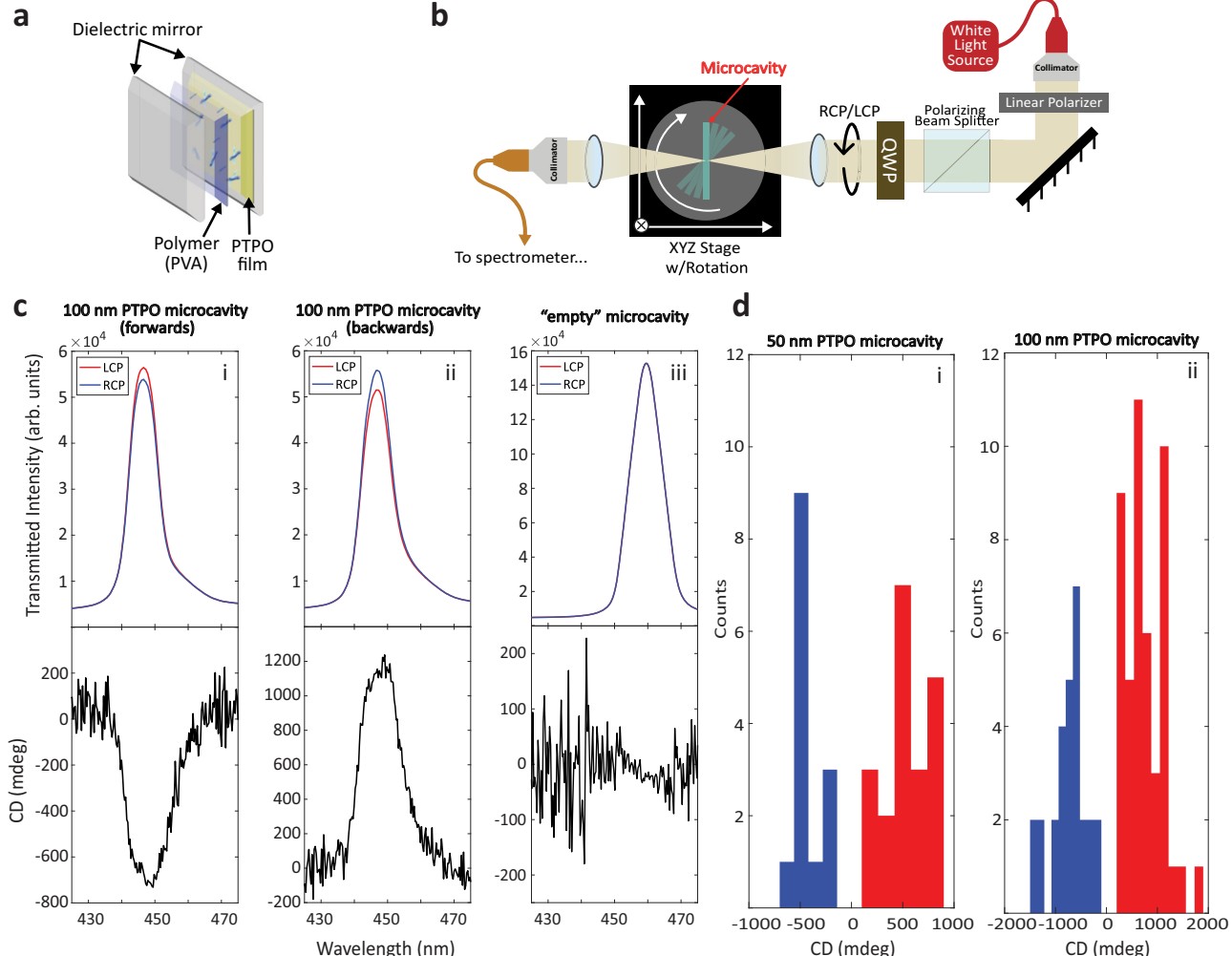

**Fig. 2 | 2D Chiral microcavity configuration and demonstration of microcavity chiroptical response. a** A schematic illustration of the FP microcavity architecture used in our experiments. **b** A single-beam white light system that directs the cavity transmission signal to a spectrometer. The quarter-wave plate (QWP) generates RCP and LCP. A rotational stage is employed to provide angular dispersion of the microcavity spectrum. **c** Transmitted intensities of RCP (red) and LCP (blue), on top, and calculated CD in mdeg (black) for the same spectral regions. (i, ii) are the same approximate spot in the same PTPO microcavity irradiated from opposite directions, while (iii) is a microcavity with no PTPO. **d** Maximum CD values measured over the resonant wavelength range of 400–500 nm for a variety of microcavities and microcavity positions for (i) 50 nm thick PTPO films and (ii) 100 nm thick PTPO films.

response of a microcavity by embedding a self-assembled chiral thin film[22], (*S,S*)-PTPO, 3,3′-(2,5-bis(((*S*)-3,7- dimethyloctyl)oxy)-1,4-pheny-lene)bis(1-(thiophen-2-yl)prop-2-yn-1-one), Fig. 1a–c. PTPO films exhibit ACD, as evidenced by the sign flip in CD as a function of sample orientation, Fig. 1d, e. As a result of the self-assembly process, the thin films are polycrystalline; therefore, all chiroptical signals are potentially composed of contributions from multiple domains oriented at different angles. To form a microcavity, a chiral thin film of PTPO and a transparent polymer binding layer (polyvinyl acetate, PVA) are embedded between two highly reflective (HR) mirrors, Fig. 2a. Importantly, not only does this structure involve a broken symmetry between LCP and RCP modes, the chiroptical signal at the cavity resonance is an order of magnitude larger than that of the thin film outside of the cavity. This phenomenon is the result of ACD signals being amplified upon experiencing an increased optical pathlength within the cavity environment, resulting in a 2D chiral microcavity.

## Results

### Characterization of thin films exhibiting ACD

The CD spectra were obtained using a white light source and a home-built broadband spectroscopy platform where the LCP and RCP are generated by a quarter-wave plate with two different angles to the fast-axis, as depicted in Fig. 2b. Illumination was provided by the collimated output from a stabilized tungsten-halogen light source, which was focused on a target for measuring the transmission spectra, see "Methods". For every measurement, the background spectra of LCP and RCP with no sample was verified to be equal, ensuring that the signal was solely due to the sample. CD spectra were recorded on the same films in forward and backward configuration. CD, measured in millidegrees (mdeg), is defined as[26–28]

$$CD = 32980 \times \log_{10} \frac{I_R}{I_L} \tag{1}$$

where $I_{R(L)}$ denotes the intensity of transmitted RCP (LCP) light. We note that CD is typically defined this way in the chemistry community (where PTPO was first described), while there are variations in defining CD in the metamaterials community[29], such as restricting it to differences in the diagonal transmission matrix elements in a circular basis. For this work, we adopt the chemistry community's definition (Eq. 1). Equivalently, our measurement is proportional to the upper right element of the Mueller matrix ($M_{03}$)[30].

To inspect the asymmetric transmission of the PTPO, thin films were deposited on a HR substrate with 95% reflectivity (Evaporated Coatings Inc., Fig. S1), the same that will ultimately be used to assemble the microcavity, Fig. 2a. The HR substrates were coated with three different thicknesses of PTPO: 300, 100, and 50 nm. Chiroptical signals from the 100 nm thick films were weaker than the 300 nm films, though a clear difference between LCP and RCP excitation emerges, resulting in a non-zero value of CD, Fig. 1d. The maximum CD values of 100–200 mdeg is near 455 nm. The value of CD is also opposite in the forward and backward configurations, which shows distinct evidence for the asymmetric transmission of the thin film and is consistent with previous measurement[22]. For 300 nm films, CD spectra are easily quantified and also show asymmetric transmission, Fig. 1e. Spatially resolved reflection-mode Mueller matrix measurements were also performed on 300 nm films, and show domains with non-zero values for $M_{03}+M_{30}$, as expected[31,32] for 2D chiral films exhibiting ACD (see Fig. S2). From comparison between CD values of the 100 and 300 nm thin films, the maximum CD value is observed to increase by roughly a factor of ten when the thin film triples in length. This behavior is qualitatively consistent with ACD's theoretical quadratic dependence on pathlength[25], though such dependence is limited by mean absorption and higher order Mueller matrix polarization effects[16]. Differences in molecular order, as further discussed below, also play a role[31]. In addition to merely having a lower signal to noise ratio than the 300 nm film, the 100 nm film acts as a less effective 2D chiral absorber per unit length. The CD spectra of the 50 nm films are too small to measure on an HR substrate without more advanced techniques. Enhancing the comparatively weak signal of the 100 and 50 nm films using microcavities in the following section represents one of the core results of this investigation.

## Experimental demonstration of a microcavity with asymmetric transmission

In PTPO films deposited on HR substrates, asymmetric transmission at the PTPO absorption band was observed at the range of 400–500 nm. It is anticipated that this feature will be amplified in a FP microcavity, where the reflectivity of the HR microcavity mirrors plays a significant role in determining the degree of temporal confinement and the linewidth of the cavity mode. These parameters are commonly expressed via the cavity finesse ($F$), which, in the limit of negligible internal cavity loss, is given by $F = -\frac{\pi}{\ln(R)}$, where $R$ corresponds to the reflectivity of the mirrors[33]. To demonstrate the cavity-enhanced chiroptic properties of the 2D chiral thin film, planar FP microcavities were fabricated. These microcavities contain the 2D chiral PTPO thin film between two mirrors and are configured as shown in Fig. 2a. The absorption spectrum of the original PTPO thin film extends below 500 nm (2.48 eV) and the spectral range of the HR mirror coating (with the reflectance over 95%) is optimized between 440 and 500 nm. By adjusting the PVA thickness[34], the resonance peak was optimized and tuned to occur within the high reflectivity window.

Typical cavity transmission spectra and CD values from 2D chiral microcavities (HR/100 nm PTPO/160 nm PVA/HR, see details in "Methods") are shown in Fig. 2c(i, ii), and demonstrate a conspicuous difference between RCP and LCP light. In contrast, an empty microcavity with no PTPO film, but instead with a thicker PVA film (HR/260 nm PVA/HR), displays no discernible difference between RCP and LCP transmission, Fig. 2c(iii). The cavity-enhanced CD value in the 2D chiral microcavity encompasses the contribution of the chiroptical response from the PTPO thin film and the enhancement provided by the microcavity. Both quantities are expected to be affected by the strength of the thin film absorption, with the chiroptic response positively correlated and the enhancement negatively correlated to it. After amplification in the FP microcavity, the CD value magnitude varies as function of sample and spot location, reaching to nearly

1000 mdeg for 50 nm films and to nearly 2000 mdeg for 100 nm films, Fig. 2d. These values are significantly larger than the CDs of the thin film on the HR substrate, which is <200 mdeg for the 100 nm film and unquantifiable for the 50 nm film, Fig. 1d, e. Scatter exists in the CD values, indicating spatial variations of the CD of the thin film as well as the microcavity finesse, as discussed next.

Comparing Fig. 1d with Fig. 2d(ii) demonstrates that the highest CD magnitudes for 100 nm PTPO microcavities are ~10–20× higher than the values measured for thin films of comparable thickness, though this comparison is inherently qualitative since the same thin film spot cannot be measured before and after microcavity construction. These same microcavities display finesse values of 30–45 (see Fig. S3). The cavity finesse can be defined as $2\pi$ multiplied by the number of round trips before decay to $1/e$ of the initial optical intensity. Since each round trip includes two passes through the PTPO film, this simple analysis predicts enhancements of 10–14×, in qualitative agreement with the analysis above. Further analysis utilizing a side-mode resonance as an internal calibration for spot-to-spot variations of the intrinsic thin film CD value shows that the enhancement of the chiroptic response increases with finesse, which is consistent with a linear fit of the microcavity enhancement to optical pathlength (Figs. S3, S5). At optical path lengths of multiple cavity round trips, the expected theoretical pathlength dependence of CD diverges from the quadratic short pathlength regime[19] and approaches a linear regime, consistent with the empirically observed linear behavior. The clear chiroptic signals of the microcavity demonstrate two key results: that the PTPO film can generate clear asymmetric transmission in the microcavity, and that the microcavity can provide significant amplification to enhance the film's intrinsic directionally non-reciprocal properties.

## Cavity angular dispersion

For a single microcavity, one can tune the resonant wavelength by changing the incident angle of radiation and therefore the photonic wavevector in the plane of the cavity[35]. Understanding this angular dependence is critical for elucidating the mechanism of chiroptic enhancement as well as evaluating the applicability of the microcavity for creation of polaritons and other applications in QIS[36]. To adjust the resonance wavelength of the cavity, the microcavities are placed on a precision rotation and translation stage for measurements at oblique incident angles, as shown in Fig. 2b. As the angle of incidence deviates from the normal direction, the resonant peak of the cavity shifts toward lower wavelengths as anticipated from photonic angular dispersion. The cavity confines the resonant wave vectors perpendicular to the surface $\mathbf{k}_z$ while the parallel component $\mathbf{k}_{||}$ remains free, resulting in a total energy blueshift that is reminiscent of a light-line.

When tuning the resonant wavelength of the 2D chiral cavity, an intriguing trend emerges at the major microcavity resonance near 440 nm, where the CD value is higher at oblique incidence than at normal incidence, Fig. 3a, b. Multiple phenomena may contribute to this trend including properties of the film and spurious signals due to the microcavity, Fig. 3c and Fig. S5. For the film, one potential contribution is due to the shifting of resonant wavelengths into spectral regions where the ACD response of PTPO may vary. However, the ACD spectra for bare films, Fig. 1d, e, is largely constant within the accessible range of the major cavity resonance (~440–450 nm). In addition, more minor spectral features are observed at higher wavelengths, Fig. S4. The HR mirrors, which consist of distributed Bragg reflectors (DBR)s, support multiple resonances in the spectral region of ACD for PTPO, resulting in multiple modes in the cavity ensemble. Increasing incident angle also increases the pathlength through the thin film, increasing the effective interaction distance, though this effect may be counteracted by a concomitant decrease in cavity finesse. A more quantitative analysis is presented below.

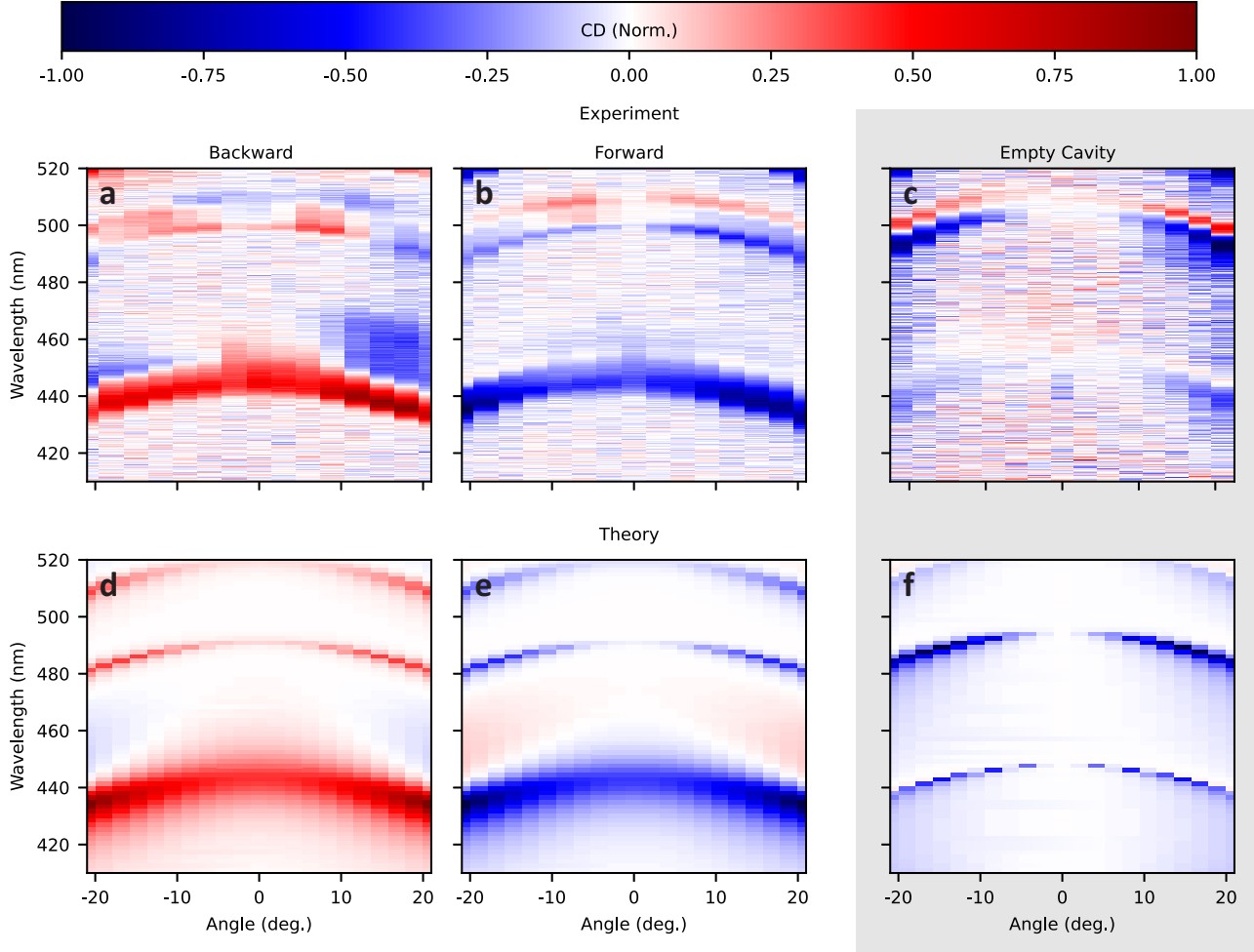

**Fig. 3 | Angular and wavelength-dependent response of CD (normalized) for microcavity containing 100 nm PTPO. a**, **b** Experimental results for forward and backward propagation and contrasted (**c**) empty cavity signals. **d**, **e** Theoretical SMM results for forward and backward propagation alongside (**f**) an empty cavity, see SI for details. Signals normalized to region of maximum magnitude of CD, which is roughly equivalent for forward and backward propagation but moderately weaker in the empty cavity.

## Theoretical model comparison with experiment

While many implementations of microcavities often contain an isotropic and/or dilute matter component between the mirrors that interacts with the light, here the matter is a thin film that is both oriented and in the solid state. As such, behaviors such as interface effects and interference within the film become relevant, necessitating an approach that properly incorporates Maxwell's equations for multiple layers of media to understand how tuning the incident light wavevector into the cavity changes the response. To quantitatively approximate the CD exhibited by the thin film, the CD signal for typical solid-state samples is expressed as follows[21]:

$$CD_{abs} = CD_{iso} + \frac{LD' \cdot LB - LD \cdot LB}{2} + LB_{res} \qquad (2)$$

Here, $CD_{iso}$ is the circular dichroism's intrinsic isotropic component which is independent of orientation, obeys Lorentz reciprocity, and does not exhibit asymmetric transmission, while LD and LB are the linear dichroism and linear birefringence. ACD produces the asymmetric transmission of RCP and LCP light due to the significant interference between linear birefringence (LB) and dichroism (LD) interactions (consequently, ACD is often referred to as LDLB). Here, we append a third term ($LB_{res}$) to allow for residual anisotropy, which stems from the inability to perform an ideal CD experiment due to

technical limitations. In typical cases, this term results from residual birefringence from the photoelastic modulator in most CD spectrometers. In our experimental setup, the third term results from a similar occurrence caused by the dielectric coating on the coverslip HR mirrors (see below and SI).

While the ACD obeys Lorentz reciprocity, its asymmetric transmission to circularly polarized light suggests some form of symmetry breaking. A system exhibiting equal transmission of light in reversed directions would possess a Jones matrix (characterizing its transmission) that is Hermitian, while the corresponding Mueller matrix would be symmetric, making symmetric transmission a significant restriction on symmetry[37]. As shown in the SI, PTPO films possess a Mueller matrix with conspicuously asymmetric contributions to CD. ACD is also intrinsically 2D chiral, due entirely to the symmetries of the sample and present at normal incidence of light, in contrast with extrinsic chirality that is due to the oblique light incidence imposing a net chirality on the system[17,38,39]. We also note that ACD should not be confused with the similarly named anisotropic CD[40,41] and also that ACD is a form of conversion circular dichroism[17] wherein the net absorption of light occurs due to conversion from LCP to RCP or vice versa.

To better understand the origins of the measured signals, we simulated angular spectra by solving Maxwell's equations within the scattering matrix method (SMM) formalism[42] (full description in SI) using an augmented version of PyLlama[43]. In such a method, we

consider light classically while using a quantum mechanical description for PTPO (see Section S7). Here, PTPO was first characterized using a combination of time-dependent density functional theory calculations alongside fitting of dipole parameters to solution and thin film spectra, Figs. S7–S10. From these calculations and subsequent empirical fitting (see Supplementary Methods S7), we determine that the electronic transition dipoles form a 2D chiral object in the $xy$ plane (Fig. 1a). Since the net collection of electronic transition dipoles is 2D chiral in aggregate, the chiroptical behavior flips upon sample flipping as is characteristic for ACD. Since we observed CD at high angles in empty microcavities and stress often manifests in DBR manufacturing[44], we also implemented a full model of the microcavity including the DBR under shear strain, Fig. S6, which accounts for the bandwidth response of the mirrors, deviations from ideal reflection in the mirrors, and the angle-dependent chiroptical response. Inclusion of this strain is necessary to reproduce the non-zero CD signal at oblique angles in Fig. 3f. While the main results of this paper address the intrinsic 2D chirality of ACD, the strained mirror and oblique light wavevector present an example of extrinsic chirality[17,38,39]. Strain breaks the symmetry of the dielectric tensor in the $xy$ plane of the DBR, but that on its own is not sufficient to produce a selective reflection of circular polarization. For that to occur, the light must hit the mirror at oblique incidence, now breaking mirror symmetry (see Section S6 for more details). We observe that the circular polarization effects are essentially identical at opposite angles in Fig. 3c, f, consistent with a manifestation of extrinsic 2D chirality where the broken symmetry is with the plane of incidence and angles of incidence $\pm\theta$ present identically as the azimuthal angle breaks the symmetry[17].

Theoretical angular-resolved spectra (Fig. 3d, e) yield excellent agreement with the experimental results for the PTPO-containing microcavities (Fig. 3a, b). This suggests that the effects of any non-idealities in the white light illumination resulting from broad-spectrum optics or other sources of artifacts are negligible, particularly in the spectral region of the fundamental cavity mode (~450 nm). The major microcavity mode (Fig. 3, major feature in all panels ~440 nm) is well-understood and reproduced theoretically. The minor peaks are highly susceptible to perturbations in microcavity mirror parameters such as mirror spacing, number of DBR periods, and DBR target wavelengths, and are only qualitatively approximated. Looking at the dispersion behavior, importantly, the increase in dissymmetry through the distinction between circularly polarized modes of light at higher angles is captured by our model.

### Spatial distribution of the CD signal

The PTPO films are not uniform samples, rather consisting of multiple crystalline domains of varying orientations[31]. Additional evidence of intrafilm diversity can be seen in time-resolved fluorescence dynamics measured at multiple film locations, Fig. S11. Each PTPO molecule consists of a centrosymmetric conjugated ring structure with two chiral oligomer side chains. As ACD behavior persists in calculations even after removing non-conjugated side chains[25], we understand the oligomers as predominantly occupying an orientational and steric role, not an electronic one. Changing the handedness of the side chain changes the sign of the CD spectra[22], which can be interpreted as being due to the preferred orientation of the molecular assembly with respect to the substrate during spin coating. According to symmetry considerations, crystalline domains oriented with the same face of their conjugated ring to the substrate show an invariant ACD, but a flipping of orientation will result in a total inversion of the sign of the ACD signal. In conventional CD spectropolarimeters, the CD signal may result from the sum of multiple reinforcing or counteracting spatial domains (referred to as grains). However, in our setup, the light is focused to a spot size of diameter ~0.24 mm that is only slightly larger than the grain size[31,45], enabling measurement of a distribution of CD signals that varies as the contributions from different grains

change. Figure 4a, b presents 2D maps of CD value for a microcavity with 100 nm PTPO versus $x$–$y$ position obtained by mapping a 11 by 11 grid array area with a 0.5 mm step size.

A diversity of CD values at different spatial locations of the microcavity are observed, Fig. 4c, including a dominant negative signal with regions of chiroptical signal inversion. This diversity is qualitatively similar to previous investigations of ACD-active oligothiophenes[22,31,45,46], but here we accomplish this imaging with thinner PTPO layers and without a high intensity synchrotron source due to the enhanced interaction length provided by the microcavity. The alternating sign and comparable length scales can also be seen in spatially resolved Mueller matrix elements of PTPO on HR substrates (see SI). The CD value consistently displays a magnitude above 300 mdeg, demonstrating that ACD exists regardless of precise grain composition. The largest magnitudes of CD nearly reach 2000 mdeg. As ACD results from the angle between microscopic LD and LB behavior for a single grain, it should be invariant to orientation of that grain within the $xy$ plane.

To understand why both signs of chiroptic response are observed, Monte Carlo simulations were performed on a modified Ising model of crystal domains (details in SI). Crystalline domains were approximated as a Voronoi tessellation[47] of the plane for which each domain could be oriented face-up or face-down, with there being a favorable interaction between adjacent domains sharing orientation that is analogous to a ferromagnetic coupling (Ising model) and an overall bias toward one face that is analogous to an applied field. We implemented such a Voronoi–Ising model to obtain a steady-state distribution at temperatures near the critical temperature[48]. In doing so, we observe two equal and opposite values for CD (Fig. 4d, yellow bars and Fig. S12) with percentages comparable to what is obtained by averaging over the experimental results (Fig. 4d, green bars). From such statistical mechanics, we support our model that PTPO molecular domains preferentially but not perfectly orient in a particular direction. Ultimately, this thermodynamic effect in the production of the thin films produces a CD response that contains regions of the dominant and of the minor polarity.

## Discussion

A variety of efforts have focused on control of chirality in cavities. For macroscopic cavities, the breaking of chiral symmetry can be realized through the use of Faraday rotators[49] or different, non-FP, cavity geometries such as ring cavities[14], methods that are not conducive for use with microcavities. Inclusion of matched quarter waveplates inside a macroscopic cavity can enable the accumulation and sensing of optical rotation, but are not compatible with use of a microcavity, while the intracavity fields are predominantly linear or helicoidal, which may not be conducive for chiral polariton formation[50,51].

Chiral plasmonic structures or metamaterials[18,52–54] can preserve circular polarization upon reflection or magnify local chiral fields, enabling construction of chiral cavities and breakthroughs in the efficient manipulation of chiral light emission[55] and interaction with chiral molecules[56,57]. Circular polarization-preserving mirrors have been fabricated at microwave[18] and visible frequencies[52]. Use of polarization-preserving mirrors allow construction of 3D chiral cavities (in other literature, simply referred to as "chiral cavities"), with several geometries having been proposed[4,58,59], and demonstrated at visible[60] and microwave frequencies[18,61]. Such 3D chiral cavities possess helical-like modes, can potentially no longer possess nodes and antinodes, and are capable of accumulating the optical rotation induced by molecules exhibiting natural chirality over multiple passes[62]. In contrast, light in a 2D chiral cavity manifests itself as angular momentum-like states, and the cavity retains the well-separated mode structure of a typical FP cavity, but is also capable of exhibiting asymmetric transmission. Importantly, a 2D chiral cavity is sufficient to form chiral polaritons[50,51].

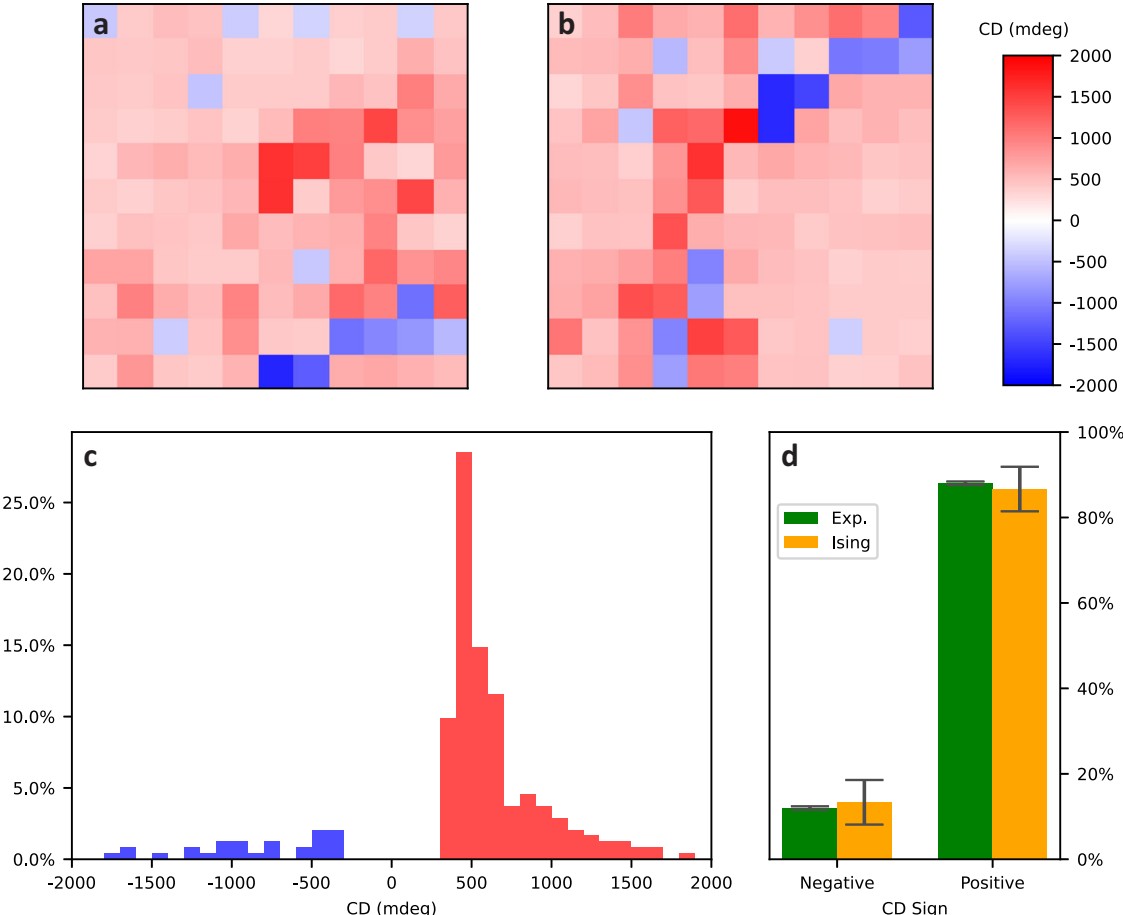

**Fig. 4 | Spatial distribution of CD for microcavity containing PTPO 100 nm thin films. a**, **b** Spatial maps of CD with step size of 0.5 mm for two non-neighboring regions of PTPO. **c** Histogram of CD for spatial maps depicted above (**a**, **b**). **d** Distribution of positive and negative CD sign and standard deviations from (**c**) compared with that for a Metropolis-Hastings simulation of an Ising model of two domain orientations. For the Ising model, CD sign given by averaging over 100 trajectories where the substrate affinity per domain area of one orientation is 16 eV mm$^{-2}$ and the interdomain coupling per shared edge length is 10 eV mm$^{-1}$ (details in SI).

Use of an organic thin film to create a 2D chiral cavity has significant advantages, but other strategies can also be applied. Asymmetric transmission of circularly polarized light can be potentially created by a 2D metamaterial that possesses an inversion center and up to $C_{2,z}$ rotational symmetry, where $z$ is the optical axis[17,29]. Confined to the plane, such systems possess an "intrinsic 2D chirality." For example, 2D chiral microscopic metamaterial structures such as crosshatches with different slit lengths can be rotated 180° out and back into the plane to become their own enantiomer[17,63]; they thereby can demonstrate asymmetric transmission of circularly polarized light. The origin of the asymmetric response in these $C_{2,z}$ metamaterials is distinct from ACD where the effect arises due to the net effect of the LD and LB axes acting as their own enantiomer when flipped, though we note that both effects can be understood as a form of conversion circular dichroism where LCP converts to RCP or vice versa[17,29]. We also note that the asymmetric response of $C_{2,z}$ metamaterial structures is distinct from that of $C_{4,z}$ structures including gammadion or gammadion-like structures that have been used to construct microcavities capable of producing circularly polarized lasing[64,65], exhibit a different Jones matrix[29] and yield mirrors that produce a phase shift, not asymmetric transmission, upon reflection[66]. Importantly, such metamaterials, whether $C_{2,z}$ or $C_{4,z}$, can prove lossy and difficult to fabricate, and to our knowledge, a 2D chiral microcavity exhibiting asymmetric transmission has not been produced with such structures.

In a recent study[67], the necessary conditions for asymmetric transmission in FP microcavities were realized based on a torsionally strained sheet illuminated at an oblique angle, which lacks the mirror symmetries of the ACD-active microcavities reported here and presents a predominantly flipped signal with respect to incident angle. In contrast, in our approach based on intrinsic 2D chirality, all angle-dependent effects are invariant with respect to the sign of the incident angle. Concerning symmetry, the Lorentz-reciprocal, non-magnetic media in our study have a response that is necessarily symmetric under flipping the sign of the angle of incidence. Said another way, in the absence of magnetic effects, dielectric behavior must be the same for incident radiation of momentum at $\mathbf{k}_x$ or $-\mathbf{k}_x$. As both strain-induced symmetric dielectric effects and ACD respect this symmetry, the optical behavior respects mirror symmetry about any plane perpendicular to the cavity. The phenomena discussed here therefore has fundamentally different symmetry considerations to the recently observed chiral cavities from Gautier, Li, Ebbesen, and Genet[67]. Importantly, their symmetry breaking predominantly affected modes at oblique incidence, complicating incorporation of such materials into photonic quantum interfaces and implementation of transduction schemes. Furthermore, we report access to a microcavity with stronger chiroptic response, particularly at normal incidence. At normal incidence, the largest CD values in our work reach 1800 mdeg, while the work of reference[67] demonstrates negligible CD values at normal incidence and a maximum signal of approximately 600 mdeg at highly oblique angles (this comparison further explained in SI). Our microcavities also demonstrate higher CD values at more oblique angles, but due to the impact of LB$_{res}$ terms at oblique angles from strain of the HR

substrate in our microcavities, we focus only the ACD-induced response at normal incidence[22].

The symmetry behavior observed in our investigation also differs with chiroptical behavior in systems such as monolayer transition-metal dichalcogenides[68] or topologically active Bloch modes[69] that invert in sign upon flipping either $\mathbf{k}_x$ or $\mathbf{k}_y$. The phenomena in ACD-active cavities should be understood in terms of the symmetry relations of reciprocal anisotropic media as opposed to gyroelectric, reciprocal magneto-electric, or non-reciprocal magneto-electric materials[70].

The results above have demonstrated a means of endowing FP microcavities with an asymmetric chiroptical response using ACD, as well as using a microcavity to amplify the weak intrinsic 2D chirality of a thin film. Importantly, a 2D chiral microcavity is easily produced by spin coating the PTPO onto a common dielectric substrate, a simple approach that allows access to 2D chiral cavities without the need for costly and unscalable electron-beam lithography. As ACD-active materials invert their chiroptic response with regards to light propagation direction, they enable symmetry breaking in a standard planar microcavity, while repeated round trips result in an enhancement of the chiroptical response by over an order of magnitude.

We reiterate that the PTPO-embedded 2D chiral microcavity contains both circular polarization modes propagating within it but that it preferentially transmits RCP in one propagation direction and LCP in the other. This geometry is distinct from previously reported 3D chiral cavities. While 2D chiral microcavities do not lend themselves to uses that require helicity-preserving modes such as chiral sensing[58,61], they offer new opportunities for chiral engineering without difficult lithography.

These new opportunities offer a variety of exciting applications. By placing molecules with specific electronic degeneracies into 2D chiral cavities, polariton ring currents have been hypothesized to exist that can enhance CD signals[71]. By analogy to traditional resonator design, the amplification of ACD-active cavities offers an alternative for chiral laser construction to cholesteric liquid crystals[72] or chiral dyes[73]. An advantage of this method for chiral lasers would be the use of optimized standard mirrors to enhance optical gain and ultimately net helicity of output pulses. As ACD-active microcavities have a chiroptical response that inverts upon sample flipping and that preferentially transmits one circular polarization, they present a new means of producing polarization-dependent optical diodes[74,75] for quantum information processing[76] and the directional manipulation of light[77]. ACD exhibits Lorentz reciprocity, therefore its usage in engineering would create reciprocal diodes[78] that distinguish between polarization modes in contrast with complete optical isolators that block both modes and require nonreciprocity such as occurs via the Faraday effect[79]. Here, the explicit directionality of ACD would enable logical operations on photonic spin states in a controllable manner.

A clear goal for future research is the achievement of strong coupling in ACD-active cavities. Recently, a cavity quantum electrodynamical theory of ACD has predicted the generation of chiral polaritons arising when ACD samples are embedded in FP cavities[16]. While the present work has not demonstrated splitting of polaritonic energies, this observation is hampered by the already broad spectral lineshapes of the PTPO. Co-incorporation of chromophores with sharper spectral features in an additional material layer offers a clear path to this goal[80], and these experiments are in progress and represent a promising direction for future work.

In this work, we have demonstrated a means of engineering asymmetric transmission of planar microcavities by incorporating a 2D chiral material demonstrating ACD. As the microcavity increases with pathlength, the repeated round trips of light enabled by the cavity finesse results in a magnification of CD values by 10–20× for the 100 nm PTPO film at normal incidence, reaching extraordinary levels of dissymmetry for organic materials. Broad access to 2D chiral microcavities exhibiting asymmetric transmission at normal incidence

in a manner that does not require complex nanofabrication will provide a powerful new photonic interface for quantum transduction.

## Methods

### 2D Chiral thin film and microcavity preparation

The 2D chiral thin films are composed of 3,3′-(2,5-bis(((S)-3,7- dimethyloctyl)oxy)-1,4-phenylene)bis(1-(thiophen-2-yl)prop-2-yn-1-one), also referred to as (S,S)-PTPO or PTPO, Fig. 1c. The synthesis of PTPO was reported in ref. 22. Highly reflective DBR mirrors, HRs, were purchased from Evaporated Coatings Inc. (Willow Grove, PA, USA) with $R = 95\%$ from 440–490 nm (see SI). These HR substrates were washed first with acetone at 40 °C, then isopropyl alcohol, then heated to 80 °C. The HR substrates were then treated with an air plasma for surface activation. PTPO films were then spin coated at 2000 rpm from dichloromethane PTPO solutions at conditions designed to create a 300 nm film (26 mg/mL), 100 nm film (3.25 mg/mL), or 50 nm film (1.2 mg/mL). Then the films underwent thermal annealing at 80 °C (compound melting point 95–97 °C).

To form a microcavity, a PVA (MW = 13,000–23,000, Sigma Aldrich) polymer adhesion layer of 100–160 nm (adjusted to ensure the cavity resonance is within the mirror coating regime) is spin coated (1700–2300 rpm, 50–60 mg/mL in chlorobenzene) on one HR using spin coating. A PTPO thin film is deposited onto a second HR substrate according to the procedure above. The second HR is placed (film side down) on top of the first HR (polymer side up) in a home-built compression tool, and lamination is consequently achieved by compression at 12 mPa in an 80 °C oven for 3 h[46].

### Experimental setup for transmission circular dichroism

A fiber-coupled broadband white light source (SLS201L, Thorlabs) is collimated and directed through a Glan-Taylor linear polarizer and superachromatic quarter-wave plate (QWP, SAQWP05M-700, Thorlabs) to generate circularly polarized light. Rotating the QWP 90° allows easy switching between RCP and LCP. The beam is then focused onto the sample via a lens with focal length 60 mm and transmitted light is recollimated and collected by a multimode fiber (FG200LEA, Thorlabs) connected to an optical spectrometer (Kymera 328i DV897ECS-BV, Andor). The spot size is measured with a CCD camera (Thorlabs, CS165CU/M).

The spectrometer records the transmission intensity for LCP and RCP excitation over a wavelength range of 380–500 nm wavelengths, then the CD is calculated for the entire spectrum according to Eq. (1). The difference between LCP and RCP is expected to be zero for achiral samples and chiral samples at non-absorbing wavelengths. The transmission spectrum for the HR mirrors at normal incidence serves as a background signal check to ensure that the transmission is equal for the LCP and RCP light. This check is performed to verify that the system is properly calibrated before taking measurements. By confirming the equality of transmission for LCP and RCP, we can ensure the accuracy of subsequent measurements and eliminate any potential systematic errors introduced by the instrument or setup.

### Procedure for angle-dependent CD measurements

Microcavity samples are carefully mounted over the pivot point of a rotation stage (PR01, Thorlabs), which is itself mounted on a translation stage used to center the sample (and pivot point) in the focus point of the beam and orthogonal to the propagation axis. The sample is rotated along an axis orthogonal to the propagation axis and LCP and RCP transmission spectra are recorded for different angles. The transmission intensities are then used to calculate the CD spectra according to Eq. (1).

### Procedure for 2D CD spatial maps

Microcavity samples are mounted on an XY translation stage in the focus point of the beam and orthogonal to the propagation axis. The

sample is translated by 0.5 mm increments in $x$ and $y$, maintaining the position along the propagation axis ($z$) and orthogonality to that axis (no angle of rotation). Transmission spectra for LCP and RCP are recorded for a $5 \times 5$ mm square and used to calculate the CD spectra according to Eq. (1).

## Data availability

Data and software related to the project can be found in the repository at https://github.com/andrewsalij/ChiralCavityFigures, which is also stored on Zenodo at https://doi.org/10.5281/zenodo.10855107.

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

## Acknowledgements

Research was primarily supported as part of the Center for Molecular Quantum Transduction, an Energy Frontier Research Center funded by U.S. Department of Energy (DOE), Office of Science, Basic Energy Sciences (BES), under Award DE-SC0021314 (microcavity fabrication, characterization, and theory). In addition, L.D.B. acknowledges support from PRIN 2017 project "CHIRALAB", grant number 20172M3K5N given by the Italian Ministry (MIUR) (PTPO synthesis), and M.S.A. acknowledges support from the Air Force Office of Scientific Research award number FA9550-23-1-0181 (apparatus for sample preparation). The authors thank Bart Kahr and Jonah Greenberg for discussions regarding the Berreman 4-by-4 matrix method and Monte Carlo simulations, respectively. We thank Dr. Nina Hong and J.A. Woollam Company for assistance with Mueller matrix spectroscopic ellipsometry measurements.

## Author contributions

Conceptualization: A.S., F.Z., G.P., L.D.B., R.G., R.T., T.C. Funding acquisition: L.D.B., M.A., M.W., R.G., R.T. Investigation: A.S., F.U., J.R., K.P., P.B., T.C. Methodology: A.D., A.S., K.P., J.R., R.G., R.T., T.C. Project administration: L.D.B., R.G., R.T. Data curation: A.S., J.R., K.P., T.C. Formal analysis: A.S., K.P., T.C. Software: A.S., K.P., T.C. Supervision: F.Z., L.D.B., L.A.A., M.A., M.W., R.G., R.T. Visualization: A.S., J.R., K.P., T.C. Writing—original draft: A.S., K.P., P.B., T.C., R.G. Writing—review & editing: A.S., F.Z., J.R., G.P., K.P., L.D.B, R.G., R.T., T.C.

## Competing interests

The authors declare no competing interests.
