## [Peer Review File · Nature Communications]

REVIEWER COMMENTS

Reviewer #1 (Remarks to the Author):

The manuscript “A Chiral Microcavity based on Apparent Circular Dichroism” by Chen et al. presents an experimental study of low-symmetry resonant optical cavities and their chiral optical response. The structures were obtained by loading ordinary Fabry-Perot (FP) cavities with quantum emitters exhibiting a spectral property called “apparent circular dichroism” (ACD for short). This experimental work is strongly related to the theoretical work by Salij et al. “Chiral polaritons based on achiral Fabry-Perot cavities using apparent circular dichroism” arXiv:2208.14461v2, where the authors develop a Hamiltonian analytical model of the system presented here.

Although this is an interesting work that contributes to the rapidly evolving field of so called chiral optical cavities and chiral polaritonics, I find many statements misleading, and the interpretation of the observed results inaccurate. I will summarize below my key criticism toward this manuscript.

1. In the introduction the authors briefly discuss chiral modes of optical cavities by stating that “FP cavities can sustain chiral modes featuring well-defined circular motion of local field amplitudes” (line 56). This goes to show that the authors confuse chirality with the spin angular momentum of light. Circular polarization correlates with handedness only in the case of a single plane wave. A standing wave confined in a properly designed cavity can have a well-defined handedness, yet possess linearly polarized fields everywhere in space (for more details, see E. Plum and N. I. Zheludev, Applied Physics Letters 106, 221901 (2015); X. Fang, K. F. MacDonald, E. Plum, and N. I. Zheludev, Scientific Reports 6, 31141 (2016)).
2. From the manuscript it is not quite clear what kind of quantum emitters do possess the property of ACD. Only from the previous works of the authors (J. Am. Chem. Soc. 143, 21519–21531 (2021), ref. 41; arXiv:2208.14461v2, ref. 67) it becomes evident that the quantum emitters should possess a complicated spectrum of electric-dipole allowed transitions with their dipoles directed in a specific manner. This origin of ACD should be more clearly explained in the introductory section of the manuscript.
3. If my understanding of the ACD phenomenon is correct, the authors should not refer to the resulting optical cavity as a “chiral” one. Violating the chiral symmetry in a quantum emitter requires the simultaneous presence of an electric AND magnetic transition dipole moments (for more information the authors could consult one of the pioneering works by Govorov et al., Nano Lett. 2010, 10, 1374). A set of electric-dipole transitions oriented at different angles is not going to render the emitter chiral. I believe the corresponding ACD feature is called “apparent” for a reason – this is not true circular dichroism.
4. The same argument applies to isolated thin films of the ACD medium: as long as the medium is not composed of truly chiral microscopic emitters this films cannot be referred to as chiral.
5. The visual and contextual quality of Figure 1 in its current form is very low. The authors should at the very least consider illustrating the internal structure of the quantum emitter responsible for the ACD phenomenon.
6. To my knowledge there is no such thing in electromagnetism as “directional reciprocity”. There is one and only one kind of reciprocity principle, which mathematically dictates the symmetry of the scattering matrix of any time-invariant linear non-magnetic system.
7. If I understand the measurements protocol correctly, the authors detect total intensity transmission amplitudes for incident LCP and RCP plane waves (I_L and I_R , respectively), and refer to the resulting

asymmetry as circular dichroism (CD). This is highly inaccurate. CD should only be used to refer to unequal RCP-to-RCP and LCP-to-LCP transmission amplitudes (for more information on the subject, see the great work by Menzel et al., Physical Review A 82, 053811 (2010)). The authors should be more accurate in defining their quantitative measure of ACD. In fact, it is easy to come with an example of a system totally lacking geometric chirality (either microscopic or macroscopic), but exhibiting unequal total intensity transmission amplitudes.

8. Two consecutive equations on lines 134 and 239 are labeled identically as (1).

9. The physics underlying Eq. (2) on line 239 could have been exemplified more clearly. What exactly are LD and LB in terms of the scattering matrix elements of the system? Is CD_{iso} of the ACD system zero due to it being completely achiral? Details like this should be discussed with more care for a submission to a high-level physical journal like Nat. Commun.

10. The following sentence from the Discussion section (line 313) illustrates very well the true nature of the system: "... the PTPO-embedded microcavity contains both circular polarization modes propagating within it but that it preferentially absorbs RCP in one propagation direction and LCP in the other". Said another way, the system examined here serves as a spin(!)-selective absorber, but has little to do with chirality. I think the authors could have a more transparent and fair description of the system if they emphasize this aspect in a more consistent way and remove claims of "chiral cavities".

To conclude, the system studied by the authors comprises an ordinary FP cavity loaded with an ensemble of molecules exhibiting so called ACD, which has nothing to do with chirality, and is the result of a complicated series of non-chiral multiple scattering events. Undoubtedly, the system as a whole does present quite a bit of new physics to investigate. However, the findings suffer critically from inaccurate, and often incorrect, interpretation, and honestly weak presentation. I cannot recommend this manuscript for publication in Nat. Commun.

Reviewer #2 (Remarks to the Author):

The manuscript deals with chiral microcavity and apparent circular dichroism.

The manuscript is interesting and contains a large number of data and information, however, in my view, it is difficult to read and it is necessary to revise completely the presentation of data. The work needs a complete revision to increase its ease of reading.

First of all the introduction contain a number of information and references much more than required to highlight the state of the art and the effectiveness of the work.

Then the discussion about ACD is not very well presented , the authors should specify the difference with the idea of " effective chirality" reported by Petronijevic et al [2021 Scientific Reports 11(1),4316]. The eq. (1) and the definition of CD, measured in mdeg, is not explained and in any case different from what presented in the ref 44.

Fig. 2 d should be better explained , the wavelength is missed .

The field distribution inside the micro cavity should be presented in order to better understand the results of the manuscript, including the case in which the angular dependence is presented.

Reviewer #3 (Remarks to the Author):

In this paper, the Authors proposed a suitable way to produce a chiral Fabry-Perot microcavity (FP). The problem is not trivial; a FB consists in coupled mirrors where light bounces forward and backward in a resonant way. Each time a chiral light beam (i.e right handed circular polarised beam RCP or left handed circular polarised beam LCP) is reflected from one of the two mirrors its chiral sign is reversed. So that, if the cavity is filled with a chiral medium (left or right handed, only one of the two back-and-forward path can be useful and the other path is detrimental. This is because a chiral medium maintain its chiral sign if looked from one side or by the opposite.

As mentioned in the text, some solutions can be round cavities, like microring resonators, where light go only 'forward' or by using bulky Faraday mirrors that exploit the magneto optic effect.

Both these solutions do not allow the realisation of a 'thin' planar cavity as the FP resonator.

The possibility studied in the presented paper, investigated a polymeric material that exhibit 'Apparent Circular Dichroism' (ACD), a feature that allows the medium to apparently show opposite handedness when looked by one side with respect the opposite site.

In order to prove the feasibility of the proposed idea, i.e. a FP microcavity with inside a layer of ACD material, different experiments (corroborated by numerical calculations) were performed.

The manuscript follows a logical organisation, starting from the characterisation of the single ACD layer in order to measure the circular dichroism (CD) when the light passes through the layer in the forward and in the backward directions. The results shown in fig.1d clearly demonstrate a good CD around 450-460 nm that inverts its sign passing from forward to backward direction.

Then a film of the ACD was deposited in the FP resonator and the rest of the cavity was filled with a transparent passive layer in order to tune the FP resonance in the proper suitable wavelength around 450 nm.

To characterise the CD of FP system with ACD material, a test with a FP cavity filled with only passive material was performed, verifying that no CD is present in that case and that the resonance was suitable tuned fig.2ciii.

The characterisation of the cavity shows an enhancement of the CD response of a factor 10, in agreement with theoretical prediction, thus demonstrating the resonance effect.

Further tests were performed measuring in details the cavities as a function of beam position and angle. The overall manuscript is convincing, well written, suitable organised and the results are, in my opinion, of high impact in the community.

By the way, I have few perplexities that can be better clarified in text as minor revisions:

1) it is mentioned that the ACD polymer, when scanned along the whole surface of the FP, can present zones with opposite chirality. This was carefully measured and reported in fig.2d and fig.4. The fact is, reasonably, attributed to inverted domain where the ACD polymer self arrange in opposite direction. The good news is that the average arrangement follow what is predicted by theory and the statistics are unbalanced towards the same sign of the CD, figure 2d.

There will be a lot of single measurements where the CD has the same sign and in principle the same sample must present CD with inverted sign, when measured backward. So, that, I'm asking why in the figures 2c-i and 2c-ii where presented measurements of opposite CD stating that the two results are from two different 100nm thick samples (in the same forward direction, because it was not differently specified).

In my opinion, for the sake of clarity, it will be better to put in fig.2c-i a good measure of a specific sample in forward direction (as I supposed it was done in that figure) and in figure 2c-ii a good measure of the SAME sample in backward direction with opposite CD.

If this change cannot be done, at least I would like a comment on this.

2) in figure 3c there are the CD measurements of the empty cavity as a function of the incidence angle and wavelength. I supposed that the empty cavity should presents zero CD. However, in the measurements are present CDs with opposite signs, in particular at normal incidence, where everything must be symmetric. Moreover in fig.3f , also the theoretical CD is different from zero (but with always the same sign). These two facts worth to be exhaustively commented.

3) this is a methodological notice: in the measurements a white light was used to shine the sample with all the wavelengths together, by using broad-spectrum waveplate and by focusing on the sample with lenses. There are in these facts some source of possible non-idealities. The achromatic waveplate should not be good for all wavelengths, the different k-vectors induced by the lens can introduce an ellipticity in the side part of the beam with respect the central part, the focal length can vary with wavelengths. By looking at the good agreement with the theoretical predictions, these effects can be negligible, but can be mentioned and commented.

Point-by-point response to all reviewer comments

Reviewer #1 (Remarks to the Author):

The manuscript “A Chiral Microcavity based on Apparent Circular Dichroism” by Chen et al. presents an experimental study of low-symmetry resonant optical cavities and their chiral optical response. The structures were obtained by loading ordinary Fabry-Perot (FP) cavities with quantum emitters exhibiting a spectral property called “apparent circular dichroism” (ACD for short). This experimental work is strongly related to the theoretical work by Salij et al. “Chiral polaritons based on achiral Fabry–Perot cavities using apparent circular dichroism” arXiv:2208.14461v2, where the authors develop a Hamiltonian analytical model of the system presented here.

Although this is an interesting work that contributes to the rapidly evolving field of so called chiral optical cavities and chiral polaritonics, I find many statements misleading, and the interpretation of the observed results inaccurate. I will summarize below my key criticism toward this manuscript.

We appreciate that Reviewer 1 finds the work to be interesting and, in the responses below, we have better clarified and contextualized our results based on Reviewer 1’s comments. One theme of Reviewer 1’s comments is to better connect our work on ACD in organic films to phenomena observed in the metamaterials community, and to make sure that our language is both precise and consistent with this community. These are excellent suggestions and have substantially strengthened the manuscript.

1. In the introduction the authors briefly discuss chiral modes of optical cavities by stating that “FP cavities can sustain chiral modes featuring well-defined circular motion of local field amplitudes” (line 56). This goes to show that the authors confuse chirality with the spin angular momentum of light. Circular polarization correlates with handedness only in the case of a single plane wave. A standing wave confined in a properly designed cavity can have a well-defined handedness, yet possess linearly polarized fields everywhere in space (for more details, see E. Plum and N. I. Zheludev, Applied Physics Letters 106, 221901 (2015); X. Fang, K. F. MacDonald, E. Plum, and N. I. Zheludev, Scientific Reports 6, 31141 (2016)).

We thank the reviewer for their focus on clarity of this concept. We had inappropriately used “chiral” and “circular-polarization” interchangeably. As such, our amended manuscript now explicitly refers to circular (or elliptical) polarization where relevant. Importantly, we have altered our usage of the term “chirality,” explicitly referring to “2D chirality,” a more precise term, where relevant. This alteration of language makes a particularly important improvement to the manuscript: by distinguishing our “2D chiral microcavities” from the previously reported “3D chiral cavities” mentioned by Reviewer 1, we make one of the sources of novelty of our work very clear: 2D chiral cavities, which are first reported here, are distinct from these other geometries that require difficult-to-produce circular polarization-preserving mirrors. We also note

that this distinction of 3D vs 2D chirality has been used by Plum and Zheludev, two authors of the papers mentioned above, for example, in <http://dx.doi.org/10.1088/1464-4258/11/7/074009>.

For example, in the introductory section, we state,

“Here, we describe a new and accessible strategy for producing planar FP microcavities that exhibit asymmetric transmission¹⁷ to circularly polarized light. The property originates in the 2D chirality of an embedded molecular thin film, and as a consequence, we refer to the cavity as a “2D chiral cavity.” We note that this 2D chiral cavity is distinct from a 3D chiral cavity produced with polarization-preserving mirrors,¹⁸ as further discussed below.”

In a later section in the discussion, we now properly contextualize our 2D chiral cavities vs 3D chiral cavities, where we make it clear that, though 3D chiral cavities offer rich physics, 2D chiral cavities are sufficient to produce 2D chiral polaritons. Later on, we also highlight that 2D chiral cavities made with ACD films are much easier to produce since they do not require nanofabrication.

“Chiral plasmonic structures or metamaterials^{18,52–54} can magnify local chiral fields, enable construction of chiral cavities, and allow breakthroughs in the efficient manipulation of chiral light emission⁵⁵ and interaction with chiral molecules.^{56,57} Circular polarization-preserving mirrors have been fabricated at microwave¹⁸ and visible frequencies.⁵² Use of polarization-preserving mirrors allow construction of 3D chiral cavities (in other literature, simply referred to as “chiral cavities”), with several geometries having been proposed,^{4,58,59} and demonstrated at visible⁶⁰ and microwave frequencies.^{18,61} Such 3D chiral cavities possess helical-like modes, can potentially no longer possess nodes and antinodes, and are capable of accumulating the optical rotation induced by molecules exhibiting natural chirality over multiple passes.⁶² In contrast, light in a 2D chiral cavity manifests itself as angular momentum-like states, and the cavity retains the well-separated mode structure of a typical FP cavity, but is also capable of exhibiting asymmetric transmission. Importantly, a 2D chiral cavity is sufficient to form 2D chiral polaritons.^{50,51}

Use of an organic thin film to create a 2D chiral cavity has significant advantages, but other strategies can also be applied. Asymmetric transmission of circularly polarized light can be potentially created by a 2D metamaterial that possesses an inversion center and up to $C_{2,z}$ rotational symmetry, where z is the optical axis.^{17,29} Confined to the plane, such systems possess an “intrinsic 2D chirality.” For example, 2D chiral microscopic metamaterial structures such as crosshatches with different slit lengths can be rotated 180 degrees out and back into the plane to become their own enantiomer;^{17,63} they thereby can demonstrate asymmetric transmission of circularly polarized light. The origin of the asymmetric response in these $C_{2,z}$ metamaterials is distinct from ACD where the effect arises due to the net effect of the LD and LB axes acting as their own enantiomer when flipped, though we note that both effects can be understood as a form of conversion circular dichroism where LCP converts to RCP or *vice versa*.^{17,29} We also note that the asymmetric response of $C_{2,z}$ metamaterial structures is distinct from that of $C_{4,z}$ structures including gammadion or gammadion-like structures that have been used to construct microcavities capable of producing circularly polarized lasing,^{64,65} exhibit a different Jones matrix²⁹ and yield mirrors that produce a phase shift, not asymmetric transmission, upon reflection.⁶⁶

Importantly, such metamaterials, whether $C_{2,z}$ or $C_{4,z}$, can prove lossy and difficult to fabricate, and to our knowledge, a 2D chiral microcavity exhibiting asymmetric transmission has not been produced with such structures.”

Finally, we have edited the specific text highlighted by Reviewer 1. This now reads:

“While FP cavities can sustain modes featuring well-defined circular motion of local field amplitudes,¹⁴ and such angular motion can in principle induce angular momentum states in non-centrosymmetric samples such as monolayer transition-metal dichalcogenides,¹⁵ standard FP cavities by themselves do not normally discriminate between circularly polarized optical modes.”

2. From the manuscript it is not quite clear what kind of quantum emitters do possess the property of ACD. Only from the previous works of the authors (J. Am. Chem. Soc. 143, 21519–21531 (2021), ref. 41; arXiv:2208.14461v2, ref. 67) it becomes evident that the quantum emitters should possess a complicated spectrum of electric-dipole allowed transitions with their dipoles directed in a specific manner. This origin of ACD should be more clearly explained in the introductory section of the manuscript.

Reviewer 1 makes an excellent suggestion. We had omitted a detailed discussion of the necessary properties for ACD quantum emitters for the sake of brevity, but we have now added sentences to the introduction to clarify this point:

“A 2D chiral cavity can be created by including materials exhibiting a property called “apparent” circular dichroism (ACD), originating from the material’s 2D chirality.¹⁹ Materials possess 2D chirality when a system embedded in a plane becomes its mirror image when flipped, potentially leading to an ACD response that displays signal inversion upon sample flipping,¹⁹ Figure 1. In contrast with natural optical activity, ACD allows for asymmetric transmission, which offers the opportunity to induce directionally dependent polarization properties and overcome the critical limitation in microcavities that prevents amplification. New options for creating 2D chiral microcavities can derive from novel materials featuring chiral π -conjugated oligothiophenes that demonstrate ACD. During the fabrication process, the oligothiophenes form thin films with a preferential orientation on a substrate as the chiral conjugated sidechains anneal and self-assemble into ordered crystals which exhibit chiroptical responses of varying sign and strength.²⁰ These films demonstrate asymmetric transmission: signal inversion of the handedness of the circularly polarized component preferentially absorbed by the two opposite faces of the sample,^{21,22} as shown in Figure 1. This inversion is the signature of ACD, which does not intrinsically originate at the molecular or microscopic level as in the case of natural optical activity, but rather originates from macroscopic ordering due to nonparallel axes of linear dichroism and linear birefringence, which occurs in oriented, low symmetry systems.^{23–25} Crystalline thin films necessarily possess the requisite orientation, and the presence in them of organic chromophores with multiple electronic transitions oriented in different directions ensures that there is a selective phase shift followed by diattenuation with a different principal axis.²⁵ While the molecules responsible for ACD-active films may have chiral chemical structures, the ACD response is

fundamentally 2D chiral, occurring due to offset optical axes whose mutual orientation changes sign upon sample flipping.”

3. If my understanding of the ACD phenomenon is correct, the authors should not refer to the resulting optical cavity as a “chiral” one. Violating the chiral symmetry in a quantum emitter requires the simultaneous presence of an electric AND magnetic transition dipole moments (for more information the authors could consult one of the pioneering works by Govorov et al., Nano Lett. 2010, 10, 1374). A set of electric-dipole transitions oriented at different angles is not going to render the emitter chiral. I believe the corresponding ACD feature is called “apparent” for a reason – this is not true circular dichroism.

The reviewer is correct in their understanding that ACD is not “true CD” and that (3D) chiral symmetry of the quantum emitter need not be violated for ACD to be present. Rather, the symmetry that is broken is that of 2D chirality of the plane of polarization. We have amended our vocabulary to explicitly refer to the phenomena and cavity as “2D chiral” to aid the reader and to distinguish the phenomenon from 3D chirality. Please also refer to our response to Reviewer 1, Comment 1.

4. The same argument applies to isolated thin films of the ACD medium: as long as the medium is not composed of truly chiral microscopic emitters this films cannot be referred to as chiral.

The films were referred to as “chiral” because they possess chiral side chains (there are (*R,R*) and (*S,S*) variants of the PTPO films discussed in the manuscript, which orient differently). That said, this usage of chirality is different from our other usage where we focus on the chiroptical effects, and we thank the reviewer for this comment.

We have added explanation to the introduction discussing the chiral oligomer side chains and omitted later references to “chiral” thin films as the previous usage was needlessly confusing.

Our edits include this clarifying sentence:

“While the molecules responsible for ACD-active films may have chiral chemical structures, the ACD response is fundamentally 2D chiral, occurring due to offset optical axes whose mutual orientation changes sign upon sample flipping.”

5. The visual and contextual quality of Figure 1 in its current form is very low. The authors should at the very least consider illustrating the internal structure of the quantum emitter responsible for the ACD phenomenon.

We thank the reviewer for this comment. We have completely revamped Figure 1 (including making sure there was no degradation in image resolution). We also took out the description of reciprocity and commercial CD data, as it was distracting. Instead, we added a functional description of the asymmetric transmission endowed by the PTPO film. We decided not to

include an image of the relationship between the LD and LB vectors since these are thoroughly described in our previous works and this detail is not central to the message in this manuscript.

Fig. 1 Chiroptical properties and signal inversion of ACD-exhibiting PTPO films. (a) Structure of PTPO. (b-c) Asymmetric transmission properties of a 2D chiral thin film in the (b) forward geometry of a PTPO film demonstrating increased transmission of LCP light over RCP light, and in a (c) backward geometry where the sign of the CD signal is reversed. (d-e) Chiroptical properties of a 100 nm (d) and 300 nm (e) PTPO film on an HR substrate measured at different spatial locations. Blue curves are acquired in the forward geometry (see b) and red curves are acquired in the backwards geometry. The thick solid curves show the average from several different spots in the backward or forward direction.

6. To my knowledge there is no such thing in electromagnetism as “directional reciprocity”. There is one and only one kind of reciprocity principle, which mathematically dictates the symmetry of the scattering matrix of any time-invariant linear non-magnetic system.

The phenomenon of asymmetric transmission imposes strict limitations on transmission matrices. However, as Reviewer 1 implies, creating a new term like “directional reciprocity” to describe this is unnecessary and distracting. Consequently, we simply described the required symmetries and avoided the neologism. One section now says,

“While the ACD obeys Lorentz reciprocity, its asymmetric transmission to circularly polarized light suggests some form of symmetry-breaking. A system exhibiting equal transmission of light in reversed directions would possess a Jones matrix (characterizing its transmission) that is Hermitian, while the corresponding Mueller matrix would be symmetric, making symmetric transmission a significant restriction on symmetry.³⁷ As shown in the SI, PTPO films possess a Mueller matrix with conspicuously asymmetric contributions to CD. ACD is also *intrinsically* 2D chiral, due entirely to the symmetries of the sample and present at normal incidence of light, in contrast with extrinsic 2D chirality that is due to a net 2D chiral object incorporating an oblique light wavevector.^{17,38,39} We also note that ACD should not be confused with the similarly-named anisotropic CD^{40,41} and also that ACD is a form of conversion circular dichroism¹⁷ wherein the net absorption of light occurs due to conversion from LCP to RCP or vice versa.”

7. If I understand the measurements protocol correctly, the authors detect total intensity transmission amplitudes for incident LCP and RCP plane waves (I_L and I_R , respectively), and refer to the resulting asymmetry as circular dichroism (CD). This is highly inaccurate. CD should only be used to refer to unequal RCP-to-RCP and LCP-to-LCP transmission amplitudes (for more information on the subject, see the great work by Menzel et al., Physical Review A 82, 053811 (2010)). The authors should be more accurate in defining their quantitative measure of ACD.

The Menzel paper referred to by Reviewer 1 provides an excellent blueprint to describe the ACD phenomenon in terms used by the metamaterials community. Our system is well-described by equation 19, which captures the behavior of a system exhibiting asymmetric transmission. From mathematical analysis, ACD can be understood as a form of conversion circular dichroism (CCD), a term that denotes the presence of the off-diagonal transmission amplitudes in the Jones matrix in equation 19. Indeed, when Menzel writes, “Moreover, from a scientific point of view, such complex MMs permit us to observe unexpected and counterintuitive effects such as asymmetric transmission for circularly [20–23]... polarized light”, these references 20-23 explicitly describe structures that are 2D (or planar) chiral and exhibit conversion CD, similar to our ACD-exhibiting PTPO film.

We agree with the reviewer that there is a distinction to be made between the diagonal and off-diagonal elements of the Jones transmission matrix (or equivalently up to a phase offset, the different elements of the Mueller matrix), but we disagree that RCP-to-LCP transmission and

vice versa is not a form of circular dichroism. When we write circular dichroism, we mean any difference between transmission of circular polarization due to a form of absorption (i.e., not reflection). This is a standard definition in the chemical community that developed PTPO. However, we see that the metamaterials community has their own definition of CD. Consequently, we have added language to clearly disambiguate the definition of CD that we are employing in the paper, while also connecting it to a quantity (the Mueller Matrix M_{03} element) that is more familiar to this community.

“CD, measured in millidegrees (mdeg), is defined as^{26–28}

$$CD = 32980 * \log_{10} \frac{I_R}{I_L} \quad (1)$$

where $I_{R(L)}$ denotes the intensity of transmitted RCP (LCP) light. We note that CD is typically defined this way in the chemistry community (where PTPO was first described), while there are variations in defining CD in the metamaterials community,²⁹ such as restricting it to differences in the diagonal transmission matrix elements in a circular basis. For this work, we adopt the chemistry community’s definition (equation 1). Equivalently, our measurement is proportional to the upper right element of the Mueller matrix (M_{03}).³⁰”

We have also added a new measurement that allows our results to better connect to the metamaterials community: we have performed Mueller matrix ellipsometry on PTPO and presented it in the SI. Specifically, we show spatial distributions of key Mueller matrix elements related to the asymmetric transmission and ACD: M_{03} , M_{30} , and $M_{03}-M_{30}$. Consequently, we believe the properties of PTPO to now be properly contextualized with the metamaterials literature. The relevant new SI section is reproduced here:

“Section S3 Mueller Matrix Characterization of PTPO Films

To more fully characterize the optical response, the Mueller matrix was acquired as a function of position. We note the connection from Jones transmission matrices to a corresponding “pure” Mueller matrix where identical information is contained up to a phase offset for coherent light transport.^{1,2}

Mueller matrix ellipsometry was performed on 300nm PTPO films deposited on an HR substrate (similar to Figure 1e). Mueller matrix elements were acquired on a JA Woollam RS2 ellipsometer equipped with focusing optics yielding a 300 μ m spot. Measurements were acquired at 20° from normal incidence at 450nm. A representative Mueller matrix from a spot near the center of the sample is shown below.

$$\begin{bmatrix} 1 & -0.337 & 0.133 & -0.044 \\ -0.351 & 0.519 & -0.003 & -0.104 \\ -0.137 & -0.017 & -0.374 & 0.125 \\ -0.053 & -0.102 & -0.114 & 0.100 \end{bmatrix}$$

In contrast, Mueller matrices acquired on HR substrates with no PTPO films showed near unity values along the diagonal and near zero values among the upper right and lower left 2x2 blocks, as expected.

Mueller matrix ellipsometry was performed on an 11x11 grid covering a 10x10 mm square area. As the M_{03} and M_{30} elements are most relevant to the ACD and asymmetric transmission properties of the material, we plot maps of M_{03} , M_{30} , $M_{03}-M_{30}$ and $M_{03}+M_{30}$.

Mueller matrix ellipsometry was performed by Nina Hong at J.A. Woollam Company.

Figure S2. Mueller matrix map for elements and combinations of elements relevant for asymmetric response to circularly polarized light and ACD.

”

In fact, it is easy to come with an example of a system totally lacking geometric chirality (either microscopic or macroscopic), but exhibiting unequal total intensity transmission amplitudes.

We agree with this statement, that many 2D chiral metamaterials are capable of displaying asymmetric transmission of circularly polarized light. The central novelty of our work is that 1) we can induce asymmetric transmission *at normal incidence* and *without* a metamaterial and thus without nanofabrication and instead, use spontaneous molecular self-assembly (ie, just easy spin coating) and 2) we use this organic layer to make a microcavity exhibiting asymmetric transmission to circularly polarized light, which to our knowledge has not been done before. Still, we agree that our message was somewhat garbled before, and Reviewer 1’s comments have helped us clarify that message. The key new text is reproduced in the response to Comment 1.

8. Two consecutive equations on lines 134 and 239 are labeled identically as (1).

Thank you, this has been fixed.

9. The physics underlying Eq. (2) on line 239 could have been exemplified more clearly. What exactly are LD and LB in terms of the scattering matrix elements of the system? Is CD_iso of the ACD system zero due to it being completely achiral? Details like this should be discussed with more care for a submission to a high-level physical journal like Nat. Commun.

For the scattering matrix of a single lamina, LD, LD', LB, and LB' are components of the transmission elements. They prove challenging to disentangle in this form, so a transformation from a 2X2 complex-valued transmission matrix (upper left quadrant of the entire scattering matrix) to a 4X4 real valued Mueller matrix helps to separate them into unique matrix elements.

For the entire macroscopic system, the four variables listed above form cross-terms that are in the same matrix element as CD, which is the reason why ACD is observed. As the focus of this paper is on the physical realization of the cavity system, for the sake of brevity we have not gone into detail of the mathematical background, which our previously cited work (current reference 25) describes in detail.

That noted, we have added more phenomenological description to the introduction of how ACD arises. Please see added text reproduced in Comment #2.

10. The following sentence from the Discussion section (line 313) illustrates very well the true nature of the system: "... the PTPO-embedded microcavity contains both circular polarization modes propagating within it but that it preferentially absorbs RCP in one propagation direction and LCP in the other". Said another way, the system examined here serves as a spin(!)-selective absorber, but has little to do with chirality. I think the authors could have a more transparent and fair description of the system if they emphasize this aspect in a more consistent way and remove claims of "chiral cavities".

As described in Comment #1 and #3, we have changed our terminology to "2D chiral cavity," a more precise description.

To conclude, the system studied by the authors comprises an ordinary FP cavity loaded with an ensemble of molecules exhibiting so called ACD, which has nothing to do with chirality, and is the result of a complicated series of non-chiral multiple scattering events. Undoubtedly, the system as a whole does present quite a bit of new physics to investigate. However, the findings suffer critically from inaccurate, and often incorrect, interpretation, and honestly weak presentation. I cannot recommend this manuscript for publication in Nat. Commun.

As described in our response to Comment #7, ACD derives from intrinsic 2D chirality, a familiar concept in the metamaterials community that is discussed by many of the authors of references

that Reviewer 1 has mentioned. With Reviewer 1's helpful suggestions, we have now refined our language to be consistent with this same community, while also making the introduction more concise. We believe that the "quite a bit of new physics" that Reviewer 1 mentions is now better able to shine though.

Reviewer #2 (Remarks to the Author):

The manuscript deals with chiral microcavity and apparent circular dichroism. The manuscript is interesting and contains a large number of data and information, however, in my view, it is difficult to read and it is necessary to revise completely the presentation of data. The work needs a complete revision to increase its ease of reading.

We thank Reviewer 2 for highlighting their interest in our manuscript, while also pointing out difficulties in its presentation. In light of these comments, we have wholly revised the manuscript. The introduction has been streamlined, terminology has been refined to be more consistent with the metamaterials community, and extraneous details have been eliminated. In particular, Figure 1 has been totally remade to better present the data.

First of all the introduction contain a number of information and references much more than required to highlight the state of the art and the effectiveness of the work.

We agree. The introduction has been made much more concise. However, contextualizing our results requires some nuanced discussion, and thus much of this section (as well as new content) has been moved to the Discussion in a new section titled, "Distinction from other Chiral Photonic Approaches".

Then the discussion about ACD is not very well presented , the authors should specify the difference with the idea of " effective chirality" reported by Petronijevic et al [2021 Scientific Reports 11(1),4316].

We understand the reviewer as referring to effective *extrinsic* chirality here. We had initially considered having a discussion in the introduction on precisely this matter but had omitted it for the sake of focus. In light of the reviewer's comments, we have added clarification and references on this precise issue as ACD is a form of *intrinsic* 2D chirality, **not** extrinsic effective chirality where an oblique wavevector of light is required. Specifically, we have added the text,

"ACD is also *intrinsically* 2D chiral, due entirely to the symmetries of the sample and present at normal incidence of light, in contrast with extrinsic 2D chirality that is due to a net 2D chiral object incorporating an oblique light wavevector.^{17,35,36"}

We have also revised the discussion of ACD, including more mechanistic details.

The eq. (1) and the definition of CD, measured in mdeg, is not explained and in any case different from what presented in the ref 44.

The reviewer is correct that Ref. 44 does not explicitly refer to the 32980 mdeg factor conversion, which arises when relating circular dichroism to ellipticity. We have added relevant textbook citations for this expression to Bengt and Norden's *Circular Dichroism and Linear Dichroism* and Barron's *Molecular Light Scattering and Optical Activity*.

Fig. 2 d should be better explained , the wavelength is missed .

We have added clarification to the caption for this figure, which is specifically for the maximum CD values observed. The wavelength of maximum of the CD signal does not vary significantly, so we did not include it in this already dense figure.

The field distribution inside the micro cavity should be presented in order to better understand the results of the manuscript, including the case in which the angular dependence is presented.

We thank the reviewer for this suggestion and have added a new figure to the supporting information (reproduced below).

Figure S14. Theoretical in-plane (xy) field intensity distributions for PTPO cavity illuminated at resonant wavelength 440 nm at normal incidence (top set) and an angle of incidence of 21 degrees in the xz plane. SiO₂ spacer included to tune theoretical cavity resonance to that of experimental cavity, mimicking the PVA spacer. Field intensities normalized to maximum of right-handed (RHP) and left-handed (LHP) illuminations. DBR field distributions omitted.

Reviewer #3 (Remarks to the Author):

In this paper, the Authors proposed a suitable way to produce a chiral Fabry-Perot microcavity (FP).

The problem is not trivial; a FB consists in coupled mirrors where light bounces forward and backward in a resonant way. Each time a chiral light beam (i.e right handed circular polarised beam RCP or left handed circular polarised beam LCP) is reflected from one of the two mirrors its chiral sign is reversed.

So that, if the cavity is filled with a chiral medium (left or right handed, only one of the two back-and-forward path can be useful and the other path is detrimental. This is because a chiral medium maintain its chiral sign if looked from one side or by the opposite.

As mentioned in the text, some solutions can be round cavities, like microring resonators, where light go only 'forward' or by using bulky Faraday mirrors that exploit the magneto-optic effect.

Both these solutions do not allow the realisation of a 'thin' planar cavity as the FP resonator.

The possibility studied in the presented paper, investigated a polymeric material that exhibit 'Apparent Circular Dichroism' (ACD), a feature that allows the medium to apparently show opposite handedness when looked by one side with respect the opposite site.

In order to prove the feasibility of the proposed idea, i.e. a FP microcavity with inside a layer of ACD material, different experiments (corroborated by numerical calculations) were performed.

The manuscript follows a logical organisation, starting from the characterisation of the single ACD layer in order to measure the circular dichroism (CD) when the light passes through the layer in the forward and in the backward directions. The results shown in fig.1d clearly demonstrate a good CD around 450-460 nm that inverts its sign passing from forward to backward direction.

Then a film of the ACD was deposited in the FP resonator and the rest of the cavity was filled with a transparent passive layer in order to tune the FP resonance in the proper suitable wavelength around 450 nm.

To characterise the CD of FP system with ACD material, a test with a FP cavity filled with only passive material was performed, verifying that no CD is present in that case and that the resonance was suitable tuned fig.2ciii.

The characterisation of the cavity shows an enhancement of the CD response of a factor 10, in agreement with theoretical prediction, thus demonstrating the resonance effect.

Further tests were performed measuring in details the cavities as a function of beam position and angle.

The overall manuscript is convincing, well written, suitable organised and the results are, in my opinion, of high impact in the community.

We thank the reviewer for their feedback and estimation of the manuscript's "high impact."

By the way, I have few perplexities that can be better clarified in text as minor revisions:

1) it is mentioned that the ACD polymer, when scanned along the whole surface of the FP, can present zones with opposite chirality. This was carefully measured and reported in fig.2d and fig.4. The fact is, reasonably, attributed to inverted domain where the ACD polymer self arrange in opposite direction.

The good news is that the average arrangement follow what is predicted by theory and the statistics are unbalanced towards the same sign of the CD, figure 2d.

There will be a lot of single measurements where the CD has the same sign and in principle the same sample must present CD with inverted sign, when measured backward. So, that, I'm asking why in the figures 2c-i and 2c-ii where presented measurements of opposite CD stating that the two results are from two different 100nm thick samples (in the same forward direction, because it was not differently specified).

In my opinion, for the sake of clarity, it will be better to put in fig.2c-i a good measure of a specific sample in forward direction (as I supposed it was done in that figure) and in figure 2c-ii a good measure of the SAME sample in backward direction with opposite CD.

If this change cannot be done, at least I would like a comment on this.

We agree with the reviewer that it would be more illustrative to show the sign change in CD response at forward and backward illumination of the same approximate spot within the same PTPO film embedded in a cavity. We have made that change to figure 2ci-ii. However, it is still important to note 1) that different domains within the same ACD polymer may still exhibit opposite CD responses under the same direction of illumination and 2), it is difficult to find the *exact* same spot in the forward and backward directions, so the CD values is not a perfect inverse.

2) in figure 3c there are the CD measurements of the empty cavity as a function of the incidence angle and wavelength. I supposed that the empty cavity should presents zero CD. However, in the measurements are present CDs with opposite signs, in particular at normal incidence, where everything must be symmetric. Moreover in fig.3f , also the theoretical CD is different from zero (but with always the same sign). These two facts worth to be exhaustively commented.

We thank the reviewer for pointing out these issues. Regarding 3c, the experimental plot, we realized that the small noisy signal at zero incidence was an experimental artifact. We have added a new version of the figure taken from the same empty resonator at a different spot, and now this small non-zero signal at normal incidence is absent, which is representative of most of the acquisitions on the empty resonator.

In 3f, the CD signal is zero at normal incidence. However, at oblique angles a chiral signal is observed due to stress in the DBR coating. The manuscript, in the "Theoretical model

comparison with experiment” section, and in the SI, in S5.3, discuss the origin of this effect, but the reviewer’s comment made us realize that we did not explicitly connect this effect back to Figure 3f. We have modified the key paragraph below (new text bolded):

“Since we observed CD at high angles in empty microcavities and stress often manifests in DBR manufacturing⁴¹ we also implemented a full model of the microcavity including the DBR under shear strain, Figure S5, which accounts for the bandwidth response of the mirrors, deviations from ideal reflection in the mirrors, and the angle-dependent chiroptical response. **Inclusion of this strain is necessary to reproduce the non-zero CD signal at oblique angles in Figure 3f.** Theoretical angular-resolved spectra (Figure 3d,e) yield excellent agreement with the experimental results for the PTPO-containing microcavities (Figure 3a,b).”

3) this is a methodological notice: in the measurements a white light was used to shine the sample with all the wavelengths together, by using broad-spectrum waveplate and by focusing on the sample with lenses. There are in these facts some source of possible non-idealities. The achromatic waveplate should not be good for all wavelengths, the different k-vectors induced by the lens can introduce an ellipticity in the side part of the beam with respect the central part, the focal length can vary with wavelengths.

By looking at the good agreement with the theoretical predictions, these effects can be negligible, but can be mentioned and commented.

We thank the reviewer for their attention to this potential source of artifacts and have added a comment to the main body of the manuscript on this point. The relevant section now says,

“Theoretical angular-resolved spectra (Figure 3d,e) yield excellent agreement with the experimental results for the PTPO-containing microcavities (Figure 3a,b). This suggests that the effects of any non-idealities in the white light illumination resulting from broad-spectrum optics or other sources of artifacts are negligible, particularly in the spectral region of the fundamental cavity mode (~450 nm).”

REVIEWER COMMENTS

Reviewer #1 (Remarks to the Author):

In this revision of their manuscript "A 2D Chiral Microcavity based on Apparent Circular Dichroism" Chen et al. have addressed technical issues raised by the referees, and improved presentation of the manuscript to some degree. Despite the improvements, I'm struggling to see this work as a publication in Nature Communications. The physics behind the observed phenomena is not presented in a way that would call consistent with the Nat. Commun. standards. The manuscript features a series of obscure equations that do not coherently uncover the underlying physical side of the system, in my opinion. Instead the manuscript creates an impression of a well-written chemistry-related work. But the physical part of the work is far from clarity, coherency, and transparency.

Besides this general impression, a number of technical comments remains on the table:

1. The authors keep using the term "circular dichroism" for the quantity that is not related to circular dichroism in any way. Total intensity transmission coefficients I_R and I_L consist of co-polarized parts t_{RR} and t_{LL} , and cross-polarized parts t_{LR} and t_{RL} . Asymmetry between t_{LR} and t_{RL} , if present, is usually called circular conversion dichroism (CCD), as the authors correctly notice. The asymmetry of the full intensity transmission coefficients is called something else, but it is definitely not "circular dichroism".
2. The visual and contextual quality of Figure 1 is still low. The authors did not illustrate the internal structure of the quantum emitter responsible for the ACD phenomenon despite the referee's recommendation. The sketch of the molecule added in Fig. 1(a) does not do a great job in explaining the structure of the underlying dipolar transitions, which plays crucial role in the observed spectral response of the system.
3. The data presented in Fig. 3(f) raises many questions. For an empty cavity having all possible planes of mirror symmetry and inversion centers I would expect absolutely no asymmetry in intensity transmission of RCP and LCP light, at normal or oblique incidence. Of course, even the empty cavity would feature non-zero circular conversions (t_{RL} and t_{LR}), but those conversions would be equal, and total intensity transmissions would of course be equal for an empty cavity. Why do we see a non-trivial asymmetry transmission (the one the authors refer to as CD, even though I disagree with this notation) for an empty cavity in Fig. 3f?

Reviewer #2 (Remarks to the Author):

The manuscript has been improved according referees requests: the manuscript can be now accepted for the publication.

Reviewer #3 (Remarks to the Author):

All the points I raised have been addressed in the revised manuscript. The document was further

enhanced by incorporating the suggestions and criticisms provided by all the reviewers. In its present form, it is, in my opinion, deemed worthy of publication.

Reviewer 1

In this revision of their manuscript “A 2D Chiral Microcavity based on Apparent Circular Dichroism” Chen et al. have addressed technical issues raised by the referees, and improved presentation of the manuscript to some degree. Despite the improvements, I’m struggling to see this work as a publication in Nature Communications. The physics behind the observed phenomena is not presented in a way that would call consistent with the Nat. Commun. standards. The manuscript features a series of obscure equations that do not coherently uncover the underlying physical side of the system, in my opinion. Instead the manuscript creates an impression of a well-written chemistry-related work. But the physical part of the work is far from clarity, coherency, and transparency.

We think of the work as a materials science work, with elements of physics and chemistry. We hope the further edits below address the Reviewer’s final concerns.

Besides this general impression, a number of technical comments remains on the table:

1. The authors keep using the term “circular dichroism” for the quantity that is not related to circular dichroism in any way. Total intensity transmission coefficients I_R and I_L consist of co-polarized parts t_{RR} and t_{LL} , and cross-polarized parts t_{LR} and t_{RL} . Asymmetry between t_{LR} and t_{RL} , if present, is usually called circular conversion dichroism (CCD), as the authors correctly notice. The asymmetry of the full intensity transmission coefficients is called something else, but it is definitely not “circular dichroism”.

As we mentioned in our previous response, different communities use different definitions of CD. We have now included a new section that shows the origin of Equation 1, our definition of CD, while also acknowledging the limits of the definition and discussing other alternative definitions. This new section is reproduced below.

Section S13 Definitions of Circular Dichroism

There exist competing definitions of the term “circular dichroism” (CD) amongst different scientific communities. One standard definition of CD denotes the differential absorption of LCP and RCP by a sample and may be reported in units of absorbance or ellipticity. These units are regularly interconverted, and such signals are frequently reported as relative measurements (i.e., g-factor).³⁵⁻³⁷ This definition of CD considers primarily the observed signal and does not necessarily account for the differential contributions to the net signal via intrinsic and extrinsic 3D and 2D chiral effects. However, the chemistry community is beginning to address observed CD with additional detail as more is learned about the contribution of anisotropic effects to chiroptical signals.³⁸

The metamaterials community distinguishes between 3D and 2D chirality through the complex circular transmission matrix, which connects the transmitted field vector and the incident field vector.³⁹ Under this definition, the difference in LCP and RCP transmission intensity in the case of CD arises from an asymmetry between the co-polarized t_{ll} and t_{rr} elements; this measure of 3D chirality is independent of the direction of light propagation. In “conversion circular dichroism” (CCD), however, the difference in LCP and RCP transmission intensity results from an asymmetry between the cross-polarized t_{lr} and t_{rl} elements; this measure of 2D chirality indicates partial conversion of the polarization state of the wave and inverts light travelling in the opposite z direction.^{40,41}

In this communication, we have chosen to utilize the former definition of circular dichroism focusing on a simple differential absorption of LCP and RCP. This is due in part to the heterogeneity of our samples (which would complicate the quantitative determination of the complex circular transmission matrix) as well as the specifics of our experimental design, which allowed only for the relative comparison between transmitted intensities of LCP and RCP and not the differential absorption which would have required both the incident and transmitted intensities of LCP and RCP.

The derivation of Equation 1 is as follows from Reference 36, beginning with the commonly accepted definition of CD in the chemistry community,

$$CD_{absorbance} = \Delta A = A_{LCP} - A_{RCP},$$

where A_i refers to the absorbance of a particular polarization. The absorbance,

$$\text{Absorbance} = -\log_{10} \left(\frac{I}{I_0} \right) = \log_{10}(I_0) - \log_{10}(I)$$

can be related to the logarithm base ten of the ratio of incident intensity (I_0) to transmitted intensity (I). Thus, CD can be recast as the log base ten of the ratio of transmitted intensities, assuming that I_0 is the same for both RCP and LCP, as experimentally verified.

$$\begin{aligned} CD_{absorbance} &= \log_{10} \left(\frac{I_0}{I_{LCP}} \right) - \log_{10} \left(\frac{I_0}{I_{RCP}} \right) \\ &= \log_{10}(I_0) - \log_{10}(I_{LCP}) - \log_{10}(I_0) + \log_{10}(I_{RCP}) \\ &= \log_{10}(I_{RCP}) - \log_{10}(I_{LCP}) \\ &= \log_{10} \left(\frac{I_{RCP}}{I_{LCP}} \right) \end{aligned}$$

Finally, CD is related to ellipticity and expressed in millidegrees, a commonly used unit,

$$CD_{\text{millidegrees}} = 32980 \times CD_{\text{absorbance}}$$

The factor of 32980 is approximately equal to,

$$= 1000 \frac{\ln(10)}{4} \frac{180}{\pi}$$

and accounts for the shift from degrees to millidegrees, the shift from base 10 to base e logarithms, and includes additional factors of 2 for an intensity to electric field conversion and from a Taylor expansion.

2. The visual and contextual quality of Figure 1 is still low. The authors did not illustrate the internal structure of the quantum emitter responsible for the ACD phenomenon despite the referee's recommendation. The sketch of the molecule added in Fig. 1(a) does not do a great job in explaining the structure of the underlying dipolar transitions, which plays crucial role in the observed spectral response of the system.

We have now amended Figure 1 to include the electronic transition dipole moments that interact to create the ACD phenomenon. While the reader is still pointed to Reference 25 for a fuller discussion of the origin (including QM calculations), we believe this new figure to be a good compromise between the reviewer's wishes for more insight on the molecular origin of ACD and our desire to keep the emphasis of this current manuscript on the 2D chiral microcavity. The new figure is reproduced here along with new text in the manuscript:

“When the projection of multiple electronic transition dipoles (e.g., in Figure 1a) in crystals overlap in energy while forming a 2D chiral object, ACD occurs near the electronic absorption band.²⁵”

“From these calculations and subsequent empirical fitting (see Supplementary Methods S7), we determine that the electronic transition dipoles form a 2D chiral object in the xy plane (Figure 1a). Since the net collection of electronic transition dipoles is 2D chiral in aggregate, the chiroptical behavior flips upon sample flipping as is characteristic for ACD.”

3. The data presented in Fig. 3(f) raises many questions. For an empty cavity having all possible planes of mirror symmetry and inversion centers I would expect absolutely no asymmetry in intensity transmission of RCP and LCP light, at normal or oblique incidence. Of course, even the empty cavity would feature non-zero circular conversions (t_{RL} and t_{LR}), but those conversions would be equal, and total intensity transmissions would of course be equal for an empty cavity. Why do we see a non-trivial asymmetry transmission (the one the authors refer to as CD, even though I disagree with this notation) for an empty cavity in Fig. 3f?

While we included a detailed discussion of this topic in the SI (Section S6 Analysis of CD response of mechanically strained microcavity), we did not do a good enough job of pointing the reader to this section in the main manuscript. We have added the text below to summarize the results of the calculation and point the reader to more detail in the SI.

“While the main results of this paper address the intrinsic 2D chirality of ACD, the strained mirror and oblique light wavevector present an example of *extrinsic* chirality^{17,38,39}. Strain breaks the symmetry of the dielectric tensor in the xy -plane of the DBR, but that on its own is not sufficient to produce a selective reflection of circular polarization. For that to occur, the light must hit the mirror at oblique incidence, now breaking mirror symmetry (see Section S6 for more details). We observe that the circular polarization effects are essentially identical at opposite angles in Figure 3c,f, consistent with a manifestation of “extrinsic 2D chirality” where the broken symmetry is with the plane of incidence and $\pm\theta$ present identically as the azimuthal angle breaks the symmetry¹⁷. ”

REVIEWERS' COMMENTS

Reviewer #1 (Remarks to the Author):

In the second revision of their manuscript "A 2D Chiral Microcavity based on Apparent Circular Dichroism" Chen et al. have addressed the remaining referee's comments and improved the technical side of their work.

In principle, I have no further technical question to this manuscript.

Whether this manuscript is suitable for Nat. Commun. is still very debatable. In my humble opinion, the amount of novelty and innovation presented here just does not reach the level of Nat. Commun. standards.

Reviewer #3 (Remarks to the Author):

All the points I raised have been addressed in the revised manuscript. The document was further enhanced by incorporating the suggestions and criticisms provided by all the reviewers. In its present form, it is, in my opinion, deemed worthy of publication.